# Development of Indications for Endoscopic Spine Surgery: An Overview

**Fernanda Wirth [1,\*], Esthael Cristina Querido Avelar Bergamaschi [1,2], Fábio da Silva Forti [1] and João Paulo Machado Bergamaschi [1,2]**

[1] Atualli Academy, 2504 Brigadeiro Luís Antônio, Cj. 172, São Paulo 01402-000, SP, Brazil; esthael_avelar@hotmail.com (E.C.Q.A.B.); fabio.forti@atualliacademy.com (F.d.S.F.); jberga@clinicaatualli.com.br (J.P.M.B.)

[2] Atualli Spine Care Clinic, 745 Alameda Santos, Cj. 71, São Paulo 01419-001, SP, Brazil

\* Correspondence: fwirth@gmail.com

**Abstract:** Endoscopic spine surgery (ESS) began more than 20 years ago as percutaneous endoscopic discectomy and has evolved to the present day. This technique offers many advantages, including a short hospital stay, minimal trauma and blood loss, the option of local or epidural anesthesia with sedation, a low rate of nosocomial infections, early recovery, and a quick return to work and daily activities. The success rate of this technique ranges from 83% to 90% in operated patients. This article aims to provide an overview of indications, versatility of the technique, advantages, contraindications and limitations, and also a reflection on the possible contraindications and limitations of the technique.

**Keywords:** endoscopic spine surgery; indications; evolution; development; advantages

## 1. Introduction

Endoscopic spine surgery (ESS) began as percutaneous discectomy through the efforts of Kambin and Hijikata in the 1970s. They independently introduced the posterolateral percutaneous lumbar nucleotomy technique [1,2]. At that time, decompression of the spinal canal was indirect, as the technique was guided fluoroscopically [3].

A few years later, Kambin and coworkers [4,5] developed a new method to remove the nucleus pulposus using flexible forceps and a 5 mm working cannula. Subsequently, some improvements in percutaneous endoscopic discectomy techniques were reported [6–8]. Kambin established the anatomical understanding of the triangular safe zone and transforaminal (TF) approach in 1990 [9]. This safe zone is the area where a lesion can be reached without injuring the nerves. This area is formed by the exiting nerve root, the superior endplate of the caudal vertebrae, and the superior articular process of the inferior vertebrae [9]. In this way, ESS was able to make rapid progress by using larger working channels and larger instruments [2]. In the late 1990s, Yeung [10] introduced the first fully functional endoscopic system. He used a multichannel endoscope with continuous fluid irrigation, which provided better image quality and less blood loss. They also reported successful results in cases of disc herniation [3].

With the advent of foraminoplasty in the TF approach, it was possible to remove central disc herniations and perform decompression of the lateral recess and foramen in the lumbar or thoracic spine [11]. In 1997, Osman published a cadaveric study in which the TF technique resulted in less instability and greater expansion of the foraminal space [12]. The TF approach has also shown long-term success in patients with foraminal stenosis [13,14].

With the technical development of ESS, clinical relevance became practical and standardized [15,16]. Historically, the main indication for this procedure has been soft disc herniations that may compromise a neural structure and have not been well controlled with nonsurgical therapies for a period of at least 6 weeks or show signs of worsening [17].

Most contraindications to this procedure are relative, as there are studies reporting the use of ESS in some cases that have been contraindicated in the past. The purpose of this overview is to highlight the evolution of the indications of ESS, the advantages, the versatility of the technique with the different possibilities of endoscopic approaches, the possible contraindications and limitations of the technique, and the changes related to ESS in recent years.

## 2. Development and Advantages of ESS

The development and history of ESS can be divided into four important periods. The first period is the advent of intradiscal procedures, such as percutaneous nucleotomy or percutaneous discectomy. At this time, anatomical knowledge was expanded and the first percutaneous instruments were developed. However, these were blind procedures without direct vision, only with fluoroscopy, and moreover, the indication was limited. The second period concerns the first endoscopic operations, which focused on soft tissues. This was the era of the first endoscopes that allowed direct visualization of the procedure, such as in endoscopic discectomy. The concepts of inside-out and outside-in emerged, especially in the transforaminal approach. During this period, we observed an evolution of instructions with the emergence of different types of clamps (straight, curved, angled, flexible, etc.), and there was little discussion of decompression. The third period was when ESS really consolidated on the world stage, and it includes the concept of bone surgery. This was the time when technology, endoscopes, instruments, equipment, and ESS indications evolved the most. It was the time when foraminoplasty and central decompression techniques evolved and ESS began to be used in more complex cases [3]. Finally, the fourth phase can be defined as the combination of ESS with regenerative medicine, concepts in the treatment of disc disease and even spinal fusion. In this phase, the aim was to delay the development of degenerative diseases of the spine. Changes in the patient's lifestyle, dietary changes, optimization of physical rehabilitation, nutritional supplements, hormone and/or vitamin replacement, prolotherapy, and the use of orthobiologic agents such as platelet-rich plasma (PRP), platelet-rich fibrin (PRF), bone marrow aspirate (BMA), and concentrated bone marrow aspirate (BMAC) are some of the methods used in regenerative medicine. Patients are better prepared for ESS to further optimize their recovery (concept of ground preparation) [18].

The first step is to change patients' habits to promote regenerative capacity, which is influenced by complex cellular functions and molecular processes, as they require optimal conditions [18]. The concept of ground preparation is to maximize the results by activating factors that can positively influence the regenerative capacity of the human body and eliminating the unfavorable/deteriorating factors of the patient. The focus of ground preparation is on factors that can be changed, such as lifestyle and diet, including metabolic syndrome, obesity, dysbiosis, sleep regulation, alcoholism, smoking, diet, physical activity, and the use of certain types of medications [18–20]. In the first stage of ground preparation, risk factors for low-grade systemic inflammation are investigated [18,21]. Some metabolic disorders, such as diabetes mellitus and obesity, are associated with low-grade systemic inflammation [18,22]. Some habits or metabolic disorders may affect recovery protocols. For example, obesity plays an important role in mesenchymal stem cell (MSC) differentiation and efficiency. In animal studies, obesity has been shown to negatively affect the osteogenic, adipogenic, and chondrogenic potential of MSCs [18,23]. Smoking negatively affects platelet activity by inhibiting the action of tissue plasminogen [18,24]. Alcoholic beverages have negative effects on MSCs by decreasing their activity and number and limiting their multipotency. It is known that alcohol consumption inhibits platelet aggregation. Therefore, it is recommended to avoid alcohol after treatment with PRP or other orthobiological products [18,25,26]. Sleep disturbances can impair endocrine function, reduce carbohydrate tolerance, and lead to serious health disorders [18].

In the context of ESS, PRP was initially used intradiscally for degenerative disc disease with a discogenic pain component, but some studies have shown that it can also be used in

the facet joints to treat facet pain, resulting in improvement of low back pain and functional disability [27]. PRP, due to its increased concentration of secreted growth factors, can promote an anti-inflammatory milieu that is essential for the healing process, as it can increase the metabolic activity of fibroblasts and osteoblasts [28,29]. In other words, PRP has a function in local proliferation, angiogenesis, differentiation, and settlement of local cells and stem cells. The regenerative properties are determined by differentiation and growth factors released upon platelet stimulation. They also play a role in the local production of matrix proteins, including collagen. These proteins are considered to be building blocks for the restoration of normal tissues [29]. According to some previous cohort studies, PRP accelerated spinal fusion, bone formation, and reduced pain scores in patients receiving PRP compared to the control group [29,30].

PRF provides all the clinical benefits of PRP as well as a spontaneously forming fibrin scaffold that controls clot formation, harbors stem cells and growth factors, and serves as a helpful template for tissue regeneration. This is due to the fact that it is obtained by centrifugation of whole blood without any additives. Since centrifugation is performed without anticoagulants, PRF is able to form a gelatinous clot with the fibrin matrix which contains the secretion of growth factors at the clotting site [31]. During tissue repair, recruited fibroblasts remodel this fibrin matrix and begin collagen synthesis [31,32]. The combined effect of fibroblast recruitment and growth factor secretion promotes tissue regeneration and collagen formation [31]. Indeed, PRF offers advantages over PRP, especially with regard to growth factors, as PRP has a comparatively short half-life of these compounds [31,33,34].

Because of its ease of removal and great regenerative potential, BMA is the most commonly used product in degenerative spine disease. Nowadays, attempts are made to treat the entire segment affected by the degenerative disease, i.e., the disc, bone, facet joint, nerve roots, interspinous ligament, and paravertebral muscles [35]. BMA contains progenitor cells that can differentiate into osteoblasts and other MSCs. The only difficulty is to obtain enough progenitor cells [29]. MSCs are known to have the potential to differentiate into various tissues of mesenchymal origin [18]. BMA is rich in anti-inflammatory cytokines [36,37], anti-catabolic factors [38,39], and growth factors [40–42], and presumably supports the synthesis of proteoglycans [43–45] and II-type collagen [46,47]. Because of these factors, BMA influences the proinflammatory state of the disc by modulating the inflammatory response and restoring homeostasis within the disc. BMA has been used as an adjunct to fusion with local bone graft in several studies [48–55], and one study reported fusion rates of 86% to 97% [55].

BMAC is the centrifuged BMA capable of promoting angiogenesis, and has been shown to be a novel treatment for cartilage diseases such as osteoarthritis due to its osteogenic, osteoconductive, and osteoinductive properties [18]. It is postulated that by centrifuging BMAC, the differences in density gradients contribute to an appropriate concentration of MSCs, providing a more suitable injectable volume for administration to a patient [56,57]. BMAC has anti-inflammatory, immunomodulatory, and angiogenic effects and may potentially improve tissue repair [18,58]. Since BMAC has a higher concentration of MSCs, it could accelerate the restoration of hemostasis in the disc. There are already several stem cell-based approaches for the treatment of intervertebral disc degeneration and fusion that will soon come to market [29]. The use of orthobiologics in combination with ESS seems promising, but there are still few studies in the literature on this topic.

Another possibility is the use of virtual reality (VR), augmented reality (AR), and mixed reality (MR). These technologies demonstrate the advantages of virtual simulators (ESS) and can be increasingly use to enhance surgical training programs, preoperative planning, and intraoperative applications [59].

Improvements in design and newer instrumentation certainly benefit the development of ESS. This development is primarily due to advances in novel endoscopic instruments that allow for better illumination, high-resolution imaging, and complementary technologies [60]. An example of these advances is the changes in the cutting surfaces and shapes of the endoscopic milling heads used in ESS. Technological advances have allowed us to use

ESS for many previous contraindications. With the advent of new possibilities for surgical access, its use has also been optimized. Figure 1 shows the possible endoscopic approaches to the lumbar, cervical and thoracic spine.

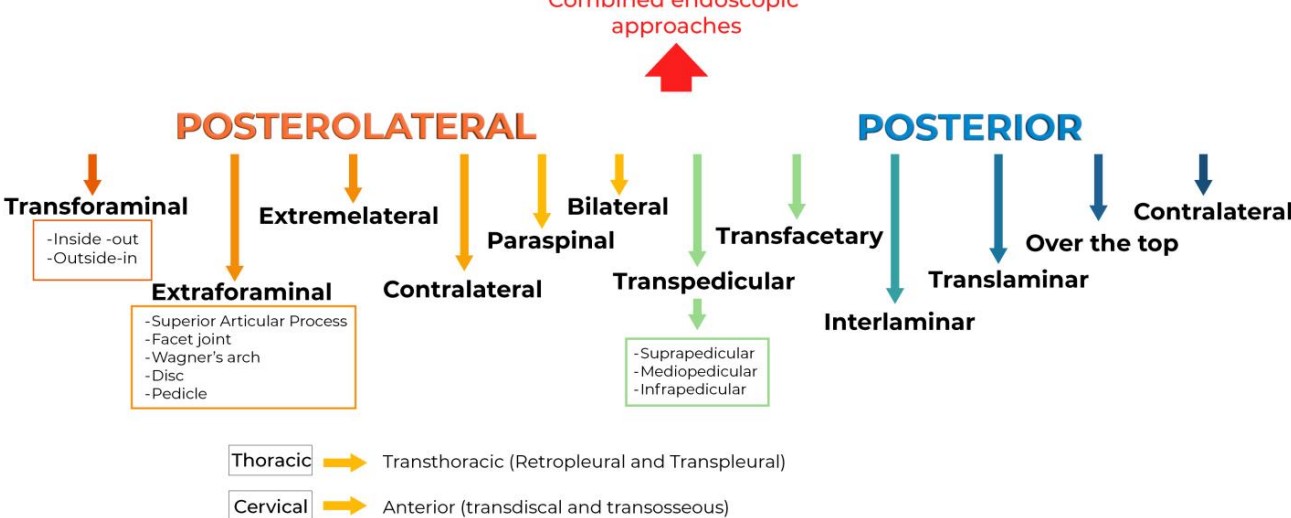

**Figure 1.** Types of endoscopic approaches.

Compared with open spine surgery, ESS has several advantages, such as a small skin incision, minimal blood loss, no muscle retraction, no excessive bone removal, minimal nerve manipulation, performed under local anesthesia and sedation, shorter operation time, and rapid return of the patient to normal daily life and activities. In addition, ESS has a very low rate of local and nosocomial infections and fewer side effects [61,62].

### 3. Complications and Contraindications of ESS

Although ESS has many advantages over open spine surgery, complications also has occur, such as incomplete decompression and incomplete removal of the intended disc fragments (these two are the most common complications), followed by inadvertent dural tear and nerve root damage. Other less common complications include persistent pain, dysesthesias, transient neuropraxia, nerve root injury, epidural hematoma, hernia recurrence, infection, postoperative instability, vascular injury, and visceral injury [17]. It is important to be aware of all possible problems that may occur during surgery, even if they are rare. It is also important to have plans in place to prevent or mitigate patient injuries [14,17,63]. Interestingly, some complications are specifically related to the use of endoscopic techniques, such as postoperative headache and postoperative seizures after prodromal neck pain. These complications are thought to be likely related to epidural pressure from endoscopic epidural irrigation, but fortunately, all of these cases are self-limiting [17,64]. Despite the advanced optical technology available nowadays, one of the main difficulties of ESS is the inability of visualization to obtain a complete image of the surgical field and adjacent structures at all times. In general, however, the risk of complications with ESS is low.

Most previous contraindications to ESS are now relative [14,17,65]. Until 2017, contraindications included: high-grade migration, calcified disc herniation (CDH), recurrent disc herniation (reoperation), more than one level, spinal stenosis and/or foraminal stenosis, spondylolisthesis, cauda equina syndrome (CES), nerve root abnormalities, and tumours [66]. With advances in devices and technology, even more contraindications associated with ESS have become relative. However, one of the remaining obstacles on the ESS website is segmental instability of the level being treated [17].

One of the most classic contraindications is CDH, as this pathology carries high risks and challenges for many reasons. CDH is defined as a subtype of disc herniation in

which the herniation site is calcified. Often, the dural sac is adhered to the annulus, and endoscopic instruments do not provide sufficient manoeuvrability to carefully remove one from the other, which could increase the risk of iatrogenic injury to the dura mater [17]. Nevertheless, ESS can currently be used safely and with satisfactory results in CDH. The limiting factors for the use of ESS in these cases are surgeon skill and learning curve [67].

Another relative contraindication is multilevel and multisectoral stenosis. Some authors hypothesize that this type of stenosis is better treated with traditional open surgery, which requires careful inspection and visualization of each level to confirm that adequate decompression has been achieved, outweighing the potential benefits of ESS [17]. With the advances of ESS and in experienced hands, this argument has become moot [68].

In cases where tumors are partially responsible for the patient´s pathology and compression symptoms, ESS requires a greater degree of skill, as the neoplasm often deforms normal structures, easily leading to disorientation and iatrogenic injury [17]. Bone tumors such as osteoid osteoma can be completely removed by ESS, according to some reports in the literature [69].

The only formal contraindication to isolated ESS is segmental instability, in which complete endoscopic discectomy or decompression may result in a transient outcome, as instability may be responsible for high rates of symptom recurrence [70]. In such cases, endoscopically assisted arthrodesis may be an option. Dissection of the disc can be performed under fluoroscopy and reviewed with the endoscope. Percutaneous or expansive cages can be used along with bone grafts, and posterior fixation with percutaneous pedicle screws or facet screws is mandatory.

## 4. What Changed in the ESS Context?

Originally, ESS was used only for lumbar discectomy, but recently, with the advancement of instruments and camera systems, it has become possible to treat many degenerative spinal conditions in the cervical and thoracic spine, so that the indications for the use of ESS have expanded considerably and continue to increase [17,71]. ESS technology has revolutionized spine surgery, as the development of high-speed drills, probes, and curved forceps has enabled the treatment of various types of disc herniations and spinal stenosis [72–74].

Nowadays, there are several case reports and studies based on previous contraindications. These studies show that these contraindications can be relative and ESS can be used with a clear conscience, since ESS can see injuries in a significantly magnified view and even the finest areas can be treated with this technique. It can be used not only for mechanical surgical injuries, but probably for many pain-causing areas that have been difficult to treat [75]. ESS has been used for decompression in degenerative spinal stenosis, disc herniation, migrated disc herniation, calcified disc herniation, scoliosis, spondylolisthesis, tumour, previous fusion, lumbar facet cysts, spinal fractures, cauda equina syndrome, and infection [70,74,76–83].

Spinal stenosis is defined as a narrowing of the spinal canal resulting in radicular impairment or clinical symptoms attributable to the spinal cord. Spinal stenosis can affect the cervical, thoracic, or lumbar spine, and can be either unilateral or bilateral and monosegmental or multisegmental [84]. Depending on the location of the stenosis, a distinction is made between central, lateral recession and foraminal stenosis. Central stenosis is most commonly found at the L4–L5 level, followed by L3–L4, L5–S1, and L1–L2 [84,85]. In lumbar spinal stenosis (LSS) without instability, some studies have supported the efficacy of ESS as a treatment option [70,86–89]. Because ESS requires minimal incision and much less soft tissue damage, and also spares the facet joints and posterior ligaments, it can lead to preservation of vertebral segment stability, unlike open surgery [90].

Migrated disc herniation (MDH) and CDH can be challenging in ESS, although successful outcomes have been reported in these cases [74,91–100]. MDH is commonly classified based on radiographic findings according to Lee et al. (2007). This classification is a schematic representation of the herniated disc and divides the direction and degree of migration into four zones (high-grade upward, low-grade upward, low-grade downward,

and high-grade downward) [101]. High-grade migrated hernias are traditionally treated with the interlaminar technique, mainly at L5–S1. Low-grade migrated hernias can be treated via a transforaminal or interlaminar approach, depending on criteria such as the level approached, associated stenosis, facet joint inclination, and surgeon preference. CDH is defined as a subtype of disc herniation in which the herniation site is calcified. Calcification may be caused by chronic inflammatory responses to the herniated disc, and usually occurs in cases more than 6 months old [99]. Several studies have reported that calcification may result from prolonged disease progression, nucleus pulposus changes, and unknown triggers such as infection and microtrauma [102–105]. In these cases, a hard disc is present that contains calcifications or ossifications in the dislocated portion of the disc herniation and is often associated with apophyseal osteophytes. This type of disc herniation may adhere to the adjacent nerve tissue [99]. It is difficult to remove the entire herniated disc by extracting part of the herniated mass and still achieve good results by ESS [99,106,107]. Until recently, ESS was thought to be poorly suited for the treatment of CDH [99], because of the increased risk of dura or root injury [97,102,104]. However, the risk of injury to neural structures exists with any type of technique used. Conventional open surgery has been widely used to treat calcified disc herniation, but it has some disadvantages, such as significant blood loss, large tissue injury, long operation time, and slow postoperative recovery, and muscle denervation and atrophy [98,108]. Other studies suggest that traditional open surgery may cause complications such as instability, infection, and chronic pain [109,110]. In this regard, ESS proves to be an efficient and safe option for the treatment of CDH with lower operative morbidity and fewer complications [97,99,111–113].

Lumbar facet cysts are a common degenerative spinal condition whose etiopathogenesis is not well understood, but they are frequently associated with degenerative facet disease, spondylolisthesis, and spinal trauma [81,114–116]. They most commonly occur in the lumbar spine, but they can also occur in the thoracic and cervical spine [81,117]. The L4–L5 level is the most predisposing level for cyst formation in the lumbar spine because this level is the most mobile level of the spine, followed by the L5–S1 level [81,118,119]. The cyst is often composed of synovial fluid extrusion, fibroblast secretion, mesenchymal cell proliferation, and collagen degeneration [120,121]. Although the open procedure has been the gold standard for the treatment of this pathology, some authors have described ESS as an efficient alternative for the treatment of facet cysts with less surgical time, less blood loss, less surgical trauma, less postoperative pain, and less hospitalization time [121–123]. With the development of new instruments, such as diamond burrs and shavers, it is possible to achieve satisfactory bone resection with advanced instrument mobility, making cyst removal technically feasible even in stenotic cases [121]. In summary, the indications for ESS have expanded to include all degenerative diseases, including facet cysts [121,124].

Another type of cyst is the disc cyst, a rare lesion that can be treated with ESS [125,126]. Sciatalgia and back pain are the most common symptoms of disc cyst. Motor deficits and hypoesthesia are the predominant signs of nerve root compression [127,128]. In most cases of symptomatic disc cyst, surgical treatment is required, but in some cases, it may regress spontaneously [127]. Disc cysts are often associated with degenerative disc disease, and ESS allows resection of bulging cysts into the spinal canal and efficient repair of the annulus tear [127].

In oncology, approximately 70% of all skeletal metastases are due to metastatic disease of the spine [129–131]. In these patients, open palliative surgery is often required to treat pain, neurologic deficits, and vertebral collapse [131]. However, open surgery is associated with a high complication rate [131,132], mainly due to postoperative wound infections caused by muscle dissection and denervation, and high blood loss [131,133–135]. For this reason, ESS may be the best technical option, as it is believed to result in earlier pain relief and less physiological impairment, allowing early optimization of function [131,136]. Some recent retrospective case series conclude that ESS is a suitable tool for the treatment of spinal tumours due to the above-mentioned advantages of this procedure and the favorable outcomes [131,137].

In the context of fractures, osteoporosis is the most common cause of spinal fractures in the elderly due to the aging population [138,139]. Osteoporosis leads to a loss of mineral content and trabecular connectivity, and these factors result in a decrease in vertebral strength [82]. An osteoporotic vertebral fracture is a fracture caused by osteoporosis resulting in compression of nerve roots in the corresponding lumbar segments and often occurs in patients with low back and leg pain [138,140]. ESS can be used in these cases if there is radicular or medullary compression due to injury to the posterior wall of the fractured vertebra [141]. In this context, another indication for ESS is the removal of cement that has extravasated into the canal or foramen causing neurologic symptoms or pain [142,143].

Finally, scoliosis is a three-dimensional deformation of the spine, and this deformation can cause LSS in idiopathic or degenerative cases. In older adults, LSS is a common problem. The permanent or intermittent pain is caused by compression of neural elements by bone, soft tissue (or both), or dynamic reinforcement, resulting in ischemia of the nerve roots [144,145]. Treatment of patients with LSS without existing instability or severe deformity can usually be achieved by decompressive surgery [70]. ESS is a suitable option for LSS cases caused by scoliosis [83]. Several authors [146–150] have already shown that this technique provides equivalent results to microsurgical or tubular techniques, but with advantages such as less tissue damage and shorter hospital stays [70,147,151–153]. With regards to scoliosis, Hasan et al. (2019) studied 45 patients with concomitant scoliosis and/or spondylolisthesis, 26 of whom underwent ESS and 19 underwent minimally invasive surgery. The authors concluded that the results were very similar in terms of functional outcomes, but ESS showed a lower complication rate [83].

## 5. Limitations of the Study

This article provides an overview of the changes associated with ESS, but further review articles and meta-analyses are needed to confirm all of these findings.

## 6. Conclusions

ESS has evolved over the past 20 years, and much has been learned about these procedures. In the field of modern spine surgery, ESS is one of the fastest growing techniques because there are fewer complications and less postoperative pain, patients return to their daily activities more quickly, and symptoms are better relieved. Due to better patient outcomes and lower medical costs, this procedure will tend to gain even more acceptance, importance, and popularity in the near future, especially because of the rapid increase in indications. The indications for ESS have expanded to include most degenerative diseases of the spine. One lesson we can learn is that if a technical improvement meets their needs and promotes improvement or standardization of a surgical technique and its outcomes, acceptance among surgeons will be high.

**Author Contributions:** Conceptualization, J.P.M.B. and F.W.; methodology, F.d.S.F.; software, F.d.S.F.; validation, F.d.S.F.; formal analysis, F.W. and J.P.M.B.; investigation of papers, F.W., E.C.Q.A.B., J.P.M.B. and F.d.S.F.; resources, F.W. and J.P.M.B.; data curation, J.P.M.B. and F.W.; writing—original draft preparation, F.W., E.C.Q.A.B. and J.P.M.B.; writing—review and editing, F.W., E.C.Q.A.B. and J.P.M.B.; visualization, F.d.S.F.; supervision, J.P.M.B. and F.W.; project administration, J.P.M.B. and F.W.; funding acquisition, F.W. and J.P.M.B. All authors have read and agreed to the published version of the manuscript.

**Funding:** This research received no external funding.

**Institutional Review Board Statement:** Not applicable.

**Informed Consent Statement:** Not applicable.

**Data Availability Statement:** The research data will be made available upon request by the corresponding author due to privacy.

**Acknowledgments:** We would like to thank Tom Almeida for helping us with the figure.

**Conflicts of Interest:** The authors declare no conflict of interest.

**Abbreviations**

ESS: Endoscopic spine surgery; TF: Transforaminal; PRF: Platelet-rich fibrin; BMA: Bone marrow aspirate; BMAC: Concentrated bone marrow aspirate; MSCs: Mesenchymal stem cells; PRP: Platelet-rich plasma; VR: Virtual reality; AR: Augmented reality; MR: Mixed reality; CDH: Calcified disc herniation; CES: Cauda equina syndrome; MDH: Migrated disc herniation; LSS: Lumbar spinal stenosis.

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
