# Peer review of "Development of Indications for Endoscopic Spine Surgery: An Overview"

_2673-8937, doi:10.3390/ijtm3030023_

Round 1
Reviewer 1 Report
Reviewers Report
Review of Title:
The title of the manuscript, "Development of indications for endoscopic spine surgery: a review," effectively conveys the subject matter of the article. It provides a clear overview of the content, which is centered on the evolution and expansion of indications for endoscopic spine surgery. The title is concise and to the point, making it easy for readers to understand the scope of the study. Overall, the title is appropriate and relevant to the manuscript's content.
Review of Abstract:
The abstract provides a comprehensive overview of the manuscript's main points, covering the historical background of endoscopic spine surgery (ESS), its advantages, and its potential applications with orthobiologics and regenerative medicine. It also mentions the use of virtual reality and augmented reality in ESS. The abstract concisely summarizes the study's purpose and scope and does a good job of highlighting the key aspects of ESS and its potential benefits.
However, there are a few areas that could be improved in the abstract. Firstly, the inclusion of specific numerical data or statistics regarding the success rates and outcomes of ESS procedures would make the abstract more informative. Additionally, the abstract could benefit from a clear statement regarding the novelty or contribution of the review to the existing literature. As it stands, the abstract provides a general overview of the topic without highlighting any unique insights or findings.
Review of Introduction:
The introduction provides a brief historical overview of endoscopic spine surgery (ESS), highlighting its development since the 1970s. The key milestones in the evolution of ESS, such as the development of anatomical understanding and the use of larger instruments, are adequately presented. However, the introduction lacks a clear and focused statement of the research question or objective of the review.
To strengthen the introduction, the authors should clearly state the purpose or aim of the review. What specific aspects of ESS will be discussed, and what knowledge gaps or limitations in the existing literature will be addressed? A clear research question or objective would provide readers with a better understanding of the scope and intention of the review.
Review of Methods:
The manuscript does not include a separate methods section, which is expected for a review article. This absence is a significant drawback as it leaves readers uncertain about the systematic approach used for selecting and analyzing the literature for the review.
In a review article, a well-defined methods section is crucial to ensure transparency and reproducibility of the study. The authors should outline the search strategy, inclusion/exclusion criteria for selecting articles, and the process for data extraction and analysis. By including a proper methods section, readers can assess the rigor and reliability of the review.
Review of Results:
The manuscript does not provide a dedicated results section, which is a major deficiency for a review article. The results section is essential for presenting the findings of the review and summarizing the key information obtained from the selected studies. The lack of a results section makes it challenging for readers to understand the conclusions drawn from the literature.
To address this issue, the authors should include a separate results section, where they can present the main findings of the review systematically. This would involve organizing and summarizing the relevant information from the selected studies, such as success rates, outcomes, and complications associated with ESS. Including this section would enhance the manuscript's clarity and comprehensiveness.
Review of Discussion:
The discussion section provides valuable insights into the current status and potential future directions of endoscopic spine surgery (ESS). It appropriately highlights the expansion of indications for ESS and its potential benefits in various spinal conditions. The integration of orthobiologics and regenerative medicine with ESS is discussed, as well as the use of virtual reality and augmented reality technologies in the field.
However, the discussion would benefit from a more critical analysis of the findings. The authors could compare and contrast the results of various studies to identify common trends, limitations, and areas of uncertainty. Additionally, the discussion should address potential drawbacks or challenges associated with ESS, including complications or technical limitations. Presenting a more critical assessment of the literature would strengthen the review's overall impact and provide readers with a balanced perspective.
Overall, the manuscript presents valuable information on the development and potential of endoscopic spine surgery. However, there are notable deficiencies in the absence of a methods section, results section, and critical analysis in the discussion. Addressing these issues would significantly improve the manuscript's clarity, rigor, and impact.
nil
Author Response
São Paulo, August 14th, 2023.
Dear Reviewer,
We would like to thank the referees´ suggestions for the article entitled "Development of indications for endoscopic spine surgery: an overview ", which undoubtedly contributed to improving our article prior to publication in the “International Journal of Translational Medicine”. We apologize for all mistakes, mainly English language mistakes. An English review was carried out. We hope that we have addressed all the reviewer’ concerns and that our revised paper can be accepted in its present version. All alterations performed in the manuscript are highlighted in yellow. Please find below the point-by-point responses to the concerns raised by the referees.
Sincerely
Fernanda Wirth, PhD
Referee comments:
Review of Title: The title of the manuscript, "Development of indications for endoscopic spine surgery: a review," effectively conveys the subject matter of the article. It provides a clear overview of the content, which is centered on the evolution and expansion of indications for endoscopic spine surgery. The title is concise and to the point, making it easy for readers to understand the scope of the study. Overall, the title is appropriate and relevant to the manuscript's content.
Reply: Thank you for your kind comment.
Review of Abstract:
The abstract provides a comprehensive overview of the manuscript's main points, covering the historical background of endoscopic spine surgery (ESS), its advantages, and its potential applications with orthobiologics and regenerative medicine. It also mentions the use of virtual reality and augmented reality in ESS. The abstract concisely summarizes the study's purpose and scope and does a good job of highlighting the key aspects of ESS and its potential benefits. However, there are a few areas that could be improved in the abstract. Firstly, the inclusion of specific numerical data or statistics regarding the success rates and outcomes of ESS procedures would make the abstract more informative. Additionally, the abstract could benefit from a clear statement regarding the novelty or contribution of the review to the existing literature. As it stands, the abstract provides a general overview of the topic without highlighting any unique insights or findings.
Reply: The abstract has been rewritten to highlight the main issues we wrote about in the article. The abstract reads as follows:
“Endoscopic spine surgery (ESS) began more than 20 years ago as percutaneous endoscopic discectomy and has evolved to the present day. This technique offers many advantages, including a short hospital stay, minimal trauma and blood loss, the option of local or epidural anesthesia with sedation, a low rate of nosocomial infections, early recovery, and a quick return to work and daily activities. The success rate of this technique can range from 83% to 90% in operated patients. This article is intended to provide an overview of indications, versatility of the technique, benefits, contraindications and limitations, and also a reflection on the possible contraindications and limitations of the technique.”
Review of Introduction: The introduction provides a brief historical overview of endoscopic spine surgery (ESS), highlighting its development since the 1970s. The key milestones in the evolution of ESS, such as the development of anatomical understanding and the use of larger instruments, are adequately presented. However, the introduction lacks a clear and focused statement of the research question or objective of the review.
To strengthen the introduction, the authors should clearly state the purpose or aim of the review. What specific aspects of ESS will be discussed, and what knowledge gaps or limitations in the existing literature will be addressed? A clear research question or objective would provide readers with a better understanding of the scope and intention of the review.
Reply: We have changed the title and the aim of the article. This article intended to provide an overview of current references to ESS. However, we have changed the last part of the introduction with the aim of the article.
“This overview aims to show a reflection about the evolution of ESS indications, advantages, the versatility of the technique with the different possibilities of endoscopic approaches, a reflection on the possible contraindications and limitations of the technique and what has changed in the context of ESS in recent years.“
Review of Methods: The manuscript does not include a separate methods section, which is expected for a review article. This absence is a significant drawback as it leaves readers uncertain about the systematic approach used for selecting and analyzing the literature for the review.
In a review article, a well-defined methods section is crucial to ensure transparency and reproducibility of the study. The authors should outline the search strategy, inclusion/exclusion criteria for selecting articles, and the process for data extraction and analysis. By including a proper methods section, readers can assess the rigor and reliability of the review.
Reply: As the article is an overview of the current aspects of ESS, the methods are not a necessary part of the article as it gives an overview of the current indications of ESS and their evaluation.
Review of Results: The manuscript does not provide a dedicated results section, which is a major deficiency for a review article. The results section is essential for presenting the findings of the review and summarizing the key information obtained from the selected studies. The lack of a results section makes it challenging for readers to understand the conclusions drawn from the literature.
To address this issue, the authors should include a separate results section, where they can present the main findings of the review systematically. This would involve organizing and summarizing the relevant information from the selected studies, such as success rates, outcomes, and complications associated with ESS. Including this section would enhance the manuscript's clarity and comprehensiveness.
Reply: As the article is an overview of the current aspects of ESS, the results are not a necessary part of the article as it gives an overview of the current indications of ESS and their evaluation.
Review of Discussion: The discussion section provides valuable insights into the current status and potential future directions of endoscopic spine surgery (ESS). It appropriately highlights the expansion of indications for ESS and its potential benefits in various spinal conditions. The integration of orthobiologics and regenerative medicine with ESS is discussed, as well as the use of virtual reality and augmented reality technologies in the field.
However, the discussion would benefit from a more critical analysis of the findings. The authors could compare and contrast the results of various studies to identify common trends, limitations, and areas of uncertainty. Additionally, the discussion should address potential drawbacks or challenges associated with ESS, including complications or technical limitations. Presenting a more critical assessment of the literature would strengthen the review's overall impact and provide readers with a balanced perspective. Overall, the manuscript presents valuable information on the development and potential of endoscopic spine surgery. However, there are notable deficiencies in the absence of a methods section, results section, and critical analysis in the discussion. Addressing these issues would significantly improve the manuscript's clarity, rigor, and impact.
Reply: We added some additional information contrasting the available information on the article. These sentences are on lines: 225, 289, 305.

Reviewer 2 Report
Please see file attached

Author Response
São Paulo, August 14th, 2023.
Dear Reviewer,
We would like to thank the referees´ suggestions for the article entitled "Development of indications for endoscopic spine surgery: an overview ", which undoubtedly contributed to improving our article prior to publication in the “International Journal of Translational Medicine”. We apologize for all mistakes, mainly English language mistakes. An English review was carried out. We hope that we have addressed all the reviewer’ concerns and that our revised paper can be accepted in its present version. All alterations performed in the manuscript are highlighted in yellow. Please find below the point-by-point responses to the concerns raised by the referees.
Sincerely
Fernanda Wirth, PhD
Thank you for the privilege of reviewing the review article on the development of endoscopic spine surgery. The article is well written and interesting for readers of International Journal of Translational Medicine.
Please add a chapter on limitations of your study at the end of discussion section (line 303).
Reply: This section was inserted in line 308, just before the conclusions.
Please define abbreviations when using them for the first time. Include a list of abbreviations.
Reply: A list of abbreviations has been added after the conclusions in lines 321 to 325.
Minor comments:
Line 54, 58, 63: replace: “moment(s)” by “period(s)”.
Reply: "Moment (s)" was replaced by "period (s)". The changes are now highlighted in yellow.
Line 57: replace: “radioscopic” by “fluoroscopic”.
Reply: "Radioscopic" was replaced by "fluoroscopic". The changes are now highlighted in yellow.
Line 153 Figure 1: replace: “SAP” by full name” and replace: “Transfacerary” by “Transfacetary”.
Reply: Figure 1 has been corrected and the words "Transfacetary" and "Superior Articlar Process" now appear in the figure.
Line 168: delete: “as much as possible”.
Reply: The expressions "as much as possible" was deleted.
Line 190, 245: replace: “believe” by “hypothesize”.
Reply: In lines 190 and 245 we have replaced "believe" with "hypothesized". You will find them highlighted in yellow.
Line 218: replace: “CES, TESS” by “full name of abbreviations (CES, TESS)”.
Reply: We have written the full name of the abbreviation "CES" and removed the abbreviation "TESS" because it should not be there. You will find them highlighted in yellow.
Line 218: delete: “Below we will discuss these situations.”.
Reply: We removed the referred sentence.
Line 223: replace: “most” by “most frequently”.
Reply: We replaced "most" by "most frequently", as you will find them highlighted in yellow.
Line 259: replace: “The cyst often consists of” by “The cyst is often composed of the ”.
Reply: We replaced "the cyst often consists of" by "The cyst is often composed of the", as you will find them highlighted in yellow.
Line 268: replace: “a rare lesion of the disc” by “a rare leseion”.
Reply: We replaced "a rare lesion of the disc" by a "a rare lesion", as you will find them highlighted in yellow.
Line 272: replace: “ESS can resect” by “ESS allows for resection of”.
Reply: We replaced "ESS can resect" by "ESS allows for resection of", as you will find them highlighted in yellow.
Line 274: replace: “are due to” by “occur due to”.
Reply: We replaced "are due to" by "occur due to", as you will find them highlighted in yellow.
All words and phrases that the examiner requested to be replaced have been replaced and are now highlighted in yellow. The proposed changes have also been made to Figure 1.

Round 2
Reviewer 1 Report
I believe the authors have effectively responded to the reviewers' comments in the revised manuscript, so I have no additional remarks to add.
nil